

# Identifying forecast uncertainties for biogenic gases in the Po valley related to model configuration in EURAD-IM during PEGASOS 2012

Annika Vogel[1,2,a] and Hendrik Elbern[1,2]

[1]Institute for Energy and Climate Research - Troposphere (IEK-8), Forschungszentrum Jülich, Germany
[2]Rhenish Institute for Environmental Research at the University of Cologne, Germany
[a]now also at: Institute for Geophysics and Meteorology, University of Cologne, Germany

**Correspondence:** A. Vogel (av@eurad.uni-koeln.de)

**Abstract.** Forecasts of biogenic trace gases in the planetary boundary layer (PBL) are highly affected by simulated emission- and transport processes. The Po region during the PEGASOS campaign in summer 2012 provides challenging, yet common conditions for simulating biogenic gases in the PBL. As a precursory step to a comprehensive model evaluation and uncertainty estimation, this study identifies and quantifies principal sources of forecasts uncertainties induced by various model configura-

tions. The investigation is based on the EURAD-IM chemistry transport model employing the MEGAN 2.1 biogenic emission module and RACM-MIM as gas phase chemistry mechanism. Isoprene and a composite of higher aldehydes are selected to demonstrate similarities and differences between these compounds. Two major sources of forecast uncertainties are identified in this study. Firstly, biogenic emissions appear to be exceptionally sensitive to land surface properties inducing total variations of local concentrations of up to one order of magnitude. Moreover, these sensitivities are found to be highly similar for

different gases and almost constant during the campaign, varying only diurnally. Secondly, the model configuration also highly influences regional flow patterns with significant effects on pollutant transport and mixing. As a result, surface concentrations of biogenic trace gases show large sensitivities to model configurations in this study. While isoprene concentrations are mainly sensitive to model configurations affecting the emission process, aldehydes show varying sensitivities related to both, biogenic emissions and transport processes. Especially in areas with small scale emission patterns, changes in the model configuration

are able to induce significantly different local concentrations. This effect was corroborated by diverging source regions of an exemplary airmass and thus applies also to non biogenic gases. The amount and complexity of sensitivities found in this study demonstrate the need to consider forecast uncertainties of chemical transport models with special focus on biogenic emissions and pollutant transport.

## 1 Introduction

One of the major challenges in tropospheric chemistry modeling is the skillful simulation of biogenic compounds, comprising emission, atmospheric transport, chemical reaction, and deposition. Being actively evolved in the formation of secondary organic aerosol (SOA, e.g., Geng et al., 2011; Shrivastava et al., 2017) and the photochemical production of tropospheric





ozone (e.g., Geng et al., 2011; Wu et al., 2015), biogenic volatile organic compounds (BVOCs) have a significant impact on trace gas and aerosol composition. However, important BVOCs like isoprene show a large spatio-temporal variability which hampers the investigation of regional to global effects (e.g., Mentel et al., 2013). As biogenic emissions dependent on various environmental conditions, forecasts of biogenic gases are exceptionally sensitive to the model setup like parameters and input fields.

Emili et al. (2016) claim that an investigation of major sources of forecast uncertainties is required before forecast errors can be estimated. Generally, atmospheric chemical forecasts are highly affected by a large number of model inputs and parameters, including the meteorological forcing. In this regard, Zhang et al. (2012b) state that more effort is needed on how meteorology and model setup affect chemical forecasts. Chemistry transport models (CTMs) are driven by numerical weather prediction (NWP) forecasts and thus inherit their errors. Different meteorological input from different NWP models or different setups of the NWP model are found to influence chemical forecasts significantly (e.g., Hu et al., 2010; Zhang et al., 2012a; Vogel et al., 2020). In addition, CTMs are sensitive to information on the earths surface type and vegetation distribution as well as emission maps (e.g., Ma and van Aardenne, 2004; Chen et al., 2014). The formulation of aerosol uptake and reactive chemistry may also serve as important components of the model configurations (e.g., Zhang et al., 2012c; Gama et al., 2019; Chen et al., 2019).

Concerning biogenic gases, highly complex dependencies of the emission process induce exceptionally large sensitivities to various model configurations including land use information and meteorological conditions (e.g. Wang et al., 2017; Henrot et al., 2017). These complex dependencies result in large differences in modeled biogenic emissions on a global scale. Focusing on climate related dependencies, Arneth et al. (2011) investigated global annual changes in modeled isoprene emissions due to climatology and vegetation input. Although the authors identified substantial effects of model input on globally averaged emissions, their results indicated increased sensitivity and complexity on smaller scales.

On regional and local scales, modeled trace gas distributions are increasingly influenced by boundary layer dynamics which controls advection and mixing of airmasses (e.g., Eder et al., 2006; Zhang et al., 2012b; Banks et al., 2016). Consequently, the large spatio-temporal gradients of observed biogenic gases renders regional forecasts highly sensitive to both, local emission- and flow patterns. Furthermore, forecast evaluation of biogenic gases is limited by exceptionally sparse observations suitable for comparison to CTM forecasts (e.g., Arneth et al., 2008; Guenther et al., 2012). This is the more the case, as available in situ observations of biogenic VOCs are hardly representative for an area larger than a few kilometers which are numerically resolved by CTMs. Ideally, hardly available free boundary layer probing under well mixed conditions may allow for reliable model evaluation.

In summer 2012, high-resolution observations of chemical distributions in the planetary boundary layer (PBL) were performed in the Po valley during the PEGASOS campaign (*Pan-European Gas-AeroSOls-climate interaction Study*, $http://pegasos.iceht.forth.gr/$). The Po valley is known to be one of the most polluted regions in Europe (e.g., Sogacheva et al., 2007; Israelevich et al., 2012; Finardi et al., 2014; Kontkanen et al., 2016; Sandrini et al., 2016). Firstly, this region is highly populated (Finardi et al., 2014) with associated high anthropogenic emissions from traffic, industry and power plants (e.g., Kontkanen et al., 2016). Emission hotspots in urban areas like Bologna and Modena are surrounded by agricultural areas with various emission characteristics. Moreover, heterogeneous and highly patchy agricultural surface patterns are likely to





remain unresolved in regional models with some kilometers grid spacing. Secondly, the topography of the Po valley impedes mixing and exchange of polluted airmasses on a regional scale (e.g., Sogacheva et al., 2007). This is mainly due to the Alps in the north and west and the Apennine Mountains in the southwest, framing the valley at three sides. The inner Po valley is characterized by a flat topography, supporting the development of nocturnal inversion layers during cloud free nights (Li et al., 2014). Thus, the Po valley is expected to provide challenging conditions with respect to local emission patterns and regional transport processes.

In the framework of the PEGASOS Po valley campaign 2010, a number of studies focus on in situ observations of aerosol composition (e.g., Wolf et al., 2015; Sullivan et al., 2016; Kontkanen et al., 2016; Rosati et al., 2016b; Bucci et al., 2018; Karnezi et al., 2018). Some findings point towards potential difficulties in simulating low level transport and biogenic gases in this region. According to Wolf et al. (2015) and Rinaldi et al. (2015), different local transport patterns affected observed aerosol composition in this region. Rosati et al. (2016a) found different aerosol properties in distinct coexisting layers within the evolving morning boundary layer. Regarding biogenic VOCs, Kaiser et al. (2015) observed low concentrations in the Po valley during the campaign. Additionally, Karnezi et al. (2018) found that the largest variation in simulated overall SOA during the campaign was induced by biogenic SOA components. Overall a dedicated quantitative assessment study on forecast uncertainties of biogenic trace gases prior to SOA- and ozone formation is as yet missing.

In this context, airborne observations from the PEGASOS campaign allow a detailed exemplary evaluation of simulated biogenic gases. However, effects of individual model configurations on forecasts should be taken into account when evaluating the forecast model and estimating related uncertainties. Therefore, this study presents a precursory step to a comprehensive evaluation and uncertainty estimation of regional forecasts of biogenic gases in the PBL. The main goal of this study is to identify and quantify principal sources of forecast uncertainties by evaluating effects of different model configurations on atmospheric chemical processes. With this approach, differences in simulated concentrations can be traced back to specific model configuration options affecting different parts of the modeling system. Focusing on biogenic gases, this study investigates different kinds of sensitivities related to model input and setup, including the configuration of the meteorological model. Other potential sources of uncertainties related to chemical conversions are not investigated in this study.

Section 2 gives an overview over the context of this study including the meteorological situation during the campaign and the selection of specific cases. The modeling system and important model configurations are described in Sect. 3. The evaluation of sources of forecast uncertainties in the Po valley during the PEGASOS campaign 2012 is performed in two steps. Firstly, effects of model configurations on biogenic emissions, pollutant transport and dry deposition are analyzed (Sect. 4). In the context of pollutant transport, calculating source areas of airmasses allows for a detailed analysis of modeled airmass history with respect to transport and mixing processes. Secondly, the effects of these model processes are analyzed in terms of differences in local trace gas concentrations (Sect. 5) which can be validated by PEGASOS observations in a follow up study.

The results focus on two biogenic gases with different dependencies on the model processes to demonstrate differences and similarities between gases. Isoprene is a directly emitted biogenic VOC with comparably short atmospheric lifetime. Aldehydes is a model variable representing a composite of oxidized BVOCs which are emitted biogenically and anthropologically, transported and dry deposited. Detailed results in Sect. 4 and Sect. 5 are given for a single case on 12 July 2012. Introduc-

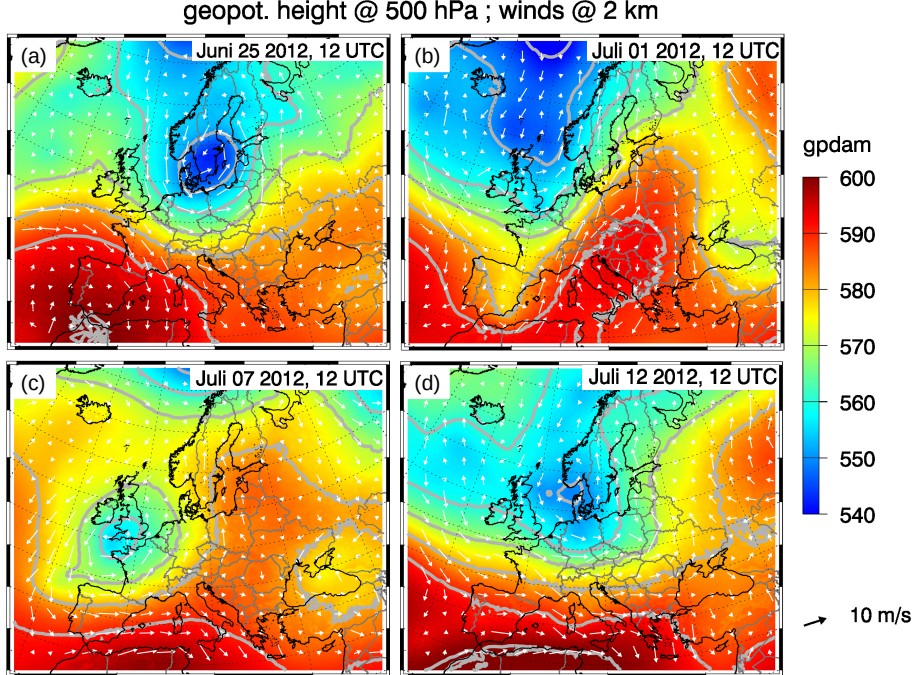

**Figure 1.** Geopotential height at 500 hPa (color coded+isolines) and horizontal winds at 2 km height (arrows) during the PEGASOS campaign in Po valley as simulated by WRF (initialized with IFS reanalysis, setup denoted as "reference" later on).

ing dominant uncertainties to the model, generalized sensitivities of biogenic emissions during the campaign are presented in Sect. 6. Finally, Sect. 7 evaluates the effects of different model configurations and concludes consequences for model evaluation 95    and uncertainty estimation.

## 2    Case Study Description

The PEGASOS campaign in the Po valley took place from 18 June to 13 July 2012. A Zeppelin NT (*New Technology*) served as airborne observational platform sampling the PBL during 22 flights over the Po valley. A detailed documentation of the Zeppelin's measurement configurations during the campaigns in 2012 is available from Jäger (2013). Section 2.1 gives a 100    short overview over the synoptic scale meteorological conditions during the campaign. The averaged daily evolution of local meteorological quantities observed at San Pietro Capofiume (SPC) during the campaign is given in Kontkanen et al. (2016). Specific cases which are identified to be representative for this study are described in Sect. 2.2.



## 2.1 Meteorological Conditions

During June and July 2012, the synoptic situation in Europe was influenced by Rossby wave activity as shown in Fig. 1.
Eastward moving troughs force several low pressure systems to move across central and northern Europe. Nevertheless, the
weather in the Po valley was continuously influenced by southern high pressure systems.

At the very beginning of the campaign, a large trough over the Atlantic ocean induced south-westerly flow over Europe. The
trough detached from the polar vortex and moved north-eastwards towards southern Sweden during the next days. On 25 June
2012, large pressure gradients between this low pressure system and a high pressure system over the Strait of Gibraltar forced
strong north-westerly to westerly winds over central Europe (Fig. 1(a) ). Under these conditions, the Alpine Mountains act
as orographic barrier resulting in low wind speeds in the leeward Po valley. During the next days, the low- and high pressure
systems weakened and moved eastwards towards the Black Sea. On 01 July 2012, a through extended from the British Islands
towards Spain, causing south-westerly flow over central Europe (Fig. 1(b) ). The trough was connected to several surface lows
and frontal activity inducing cloudy and rainy conditions from Spain to Poland and Finland. Being separated by the Alps, the
weather in northern Italy was still influenced by the weakened high pressure system in the east. Thus, clear and partly foggy
conditions with calm winds and and temperatures above $20\,^{\circ}C$ were observed in the Po valley during the morning hours on 01
July 2012.

The trough and its controlling low pressure systems propagated north-east bound during the next days. Subsequently, a cut-
off low developed south of Iceland and started to separate from the polar vortex on 04 July 2012. Three days later, on 07 July
2012, the center of the cut-off low was located over the British Islands (Fig. 1(c) ). Again, a cold front over Poland influenced
the local weather north of the Alps but did not affect the Po region. Instead, slow varying winds and clear sky conditions have
still been observed in the Po valley. These conditions in the Po valley persisted until the end of the campaign on 13 July 2012.
During this time, the cut-off low has weakened and reconnected to the polar vortex. On 12 July 2012, a new trough was formed
and moved towards the North Sea (Fig. 1(d) ) along with a sequence of associated surface lows. Additionally, a large high
pressure system formed over north-western Africa which directed westerly flow towards south-western Europe.

## 2.2 Selected Cases

The set of observational instruments onboard the Zeppelin was deployed following the emphasis assigned to each individual
PEGASOS flight (see e.g. Jäger, 2013, for an overview). Flight patterns are either vertical profiles by helical flight patterns
close to the base station at SPC or horizontal transects to different parts of the Po valley. Flight number *F049* on 12 July 2012
is an example for a helical flight pattern which was performed close to Argenta. With a radius of about 1 km, the Zeppelin
spanned altitudes from 50 m up to 750 m above ground between 03:20 and 09:20 UTC. In contrast, horizontal transects were
performed to the central Po valley, the Adriatic Sea as well as the Apennine Mountains. During some of the flights to the
Apennine Mountains – like flight number *F039* on 01 July 2012 – the Zeppelin followed a large valley towards Monte Cimone
(2165 m a.s.l.). Most of the flights took place in the early morning hours to investigate the morning development of the mixed





boundary layer. Only on 07 July 2012, flight number *F045* was scheduled between 17:00 and 20:20 UTC to investigate effects of the weakening vertical mixing during the evening hours.

This study focuses on three different cases during the PEGASOS Po valley campaign: (1) early morning on 01 July 2012, (2) evening on 07 July 2012, and (3) early morning on 12 July 2012. The selection is made upon the differences in synoptic conditions, time of day and flight pattern in order to cover different principal conditions during the campaign. Thus, these cases

allow for a generalized analysis of averaged sensitivities in the Po valley during the campaign. From these, special emphasis is placed on the early morning hours on 12 July 2012, which has already been selected for other studies (Li et al., 2014; Kaiser et al., 2015). It appears to be a typical case for investigating chemical components during the development of a morning mixed layer during this campaign.

## 3 Modeling System

The atmospheric chemical simulations are performed by the EURAD-IM modeling system, which will be introduced in Sect. 3.2. As the meteorological simulation provides an important set of model inputs, the numerical weather prediction model WRF serving as meteorological driver for EURAD-IM is shortly described in Sect. 3.1. Finally, Sect. 3.3 describes the selected model inputs and configurations which are considered during this study. For all selected cases, the simulations of WRF and EURAD-IM are initialized one day earlier at 00 UTC, each. Both models share the same domain and projection which is Lam-

bert conformal with 15 km horizontal spacing (compare Fig. 1). Based on this, a 5 km and the final 1 km domain are driven by initial- and boundary conditions from their respective mother domains. The vertical layers are defined by terrain following $\sigma$ coordinates with 23 levels extending up to 100 hPa for both models.

### 3.1 WRF numerical weather prediction

The WRF (*Weather Research and Forecasting*) model is a mesoscale numerical weather prediction model, devised by a joint
coordination effort of NCAR (National Center for Atmospheric Research), NOAA (National Oceanic and Atmospheric Administration), the U.S. Air Force, the Naval Research Laboratory, the University of Oklahoma and the Federal Aviation Administration. In this study, the advanced research WRF (WRF-ARW) version 3.8.1 is used (Skamarock et al., 2008). WRF-ARW solves a fully compressible non hydrostatic formulation of the prognostic equations. Time integration is performed by $2^{nd}$ or $3^{rd}$ order Runge-Kutta schemes. The vertical grid is defined by terrain following hydrostatic pressure coordinates and the prog-
nostic variables are horizontally staggered in an Arakawa-C-grid (Arakawa and Lamb, 1977) stencil. For a detailed description of WRF-ARW 3.8.1 see Skamarock et al. (2008).

Multiple schemes for various kinds of parameterizations are implemented in WRF-ARW. Available parameterizations account for subgrid scale processes related to the boundary- and surface layer, land- and urban surface, lake physics, short- and longwave radiation, cloud microphysics and cumulus parameterizations. Effects of these parameterizations on atmospheric
chemical forecasts are investigated in this study. The selection of parameterization schemes is given in Sect. 3.3. A detailed description of the available parameterization schemes can be found in Skamarock et al. (2008).



## 3.2 EURAD-IM chemistry transport model

The atmospheric chemical data assimilation system EURAD-IM (*EURopean Air pollution Dispersion - Inverse Model*) combines a state-of-the-art chemistry transport model with spatio-temporal data assimilation and inversion methods (Elbern et al., 2007). The chemistry transport model within EURAD-IM provides forecasts of a large set of gas phase and aerosol compounds up to lower stratospheric levels (e.g., Hass et al., 1995). Considered transformation processes include dynamical transformations due to advective and diffusive processes as well as reactive chemistry with other compounds and photolysis. For this study, the RACM-MIM (*Regional Atmospheric Chemistry Mechanism,* Geiger et al., 2003) mechanism for reactive chemistry is selected which considers 221 chemical- and 23 photolysis reactions of 84 gases including condensed isoprene degradation (*Mainz Isoprene Mechanism* MIM, Pöschl et al., 2000).

Emissions from anthropogenic- and biogenic sources are treated separately, where anthropogenic emissions are provided by the TNO-MACC-II inventory (Kuenen et al., 2014). The MEGAN 2.1 (*Model for Emissions of Gases and Aerosols from Nature version 2.1*, Guenther et al., 2012) module is used for biogenic emissions from urban, natural, and agricultural sources. In total, 147 chemical compounds are considered, which are grouped into 19 classes according to their emission properties. For each of these component classes, vegetation dependent emissions are calculated from standard emissions, multiplied by the local vegetation fraction. These vegetation dependent emissions are scaled by an activity factor to account for variations in the environmental conditions. According to Guenther et al. (2012), effects of radiation, temperature, leaf age, soil moisture, and $CO_2$ are included in a multiplicative manner. Therefore, required input parameters include fields of plant functional types (PFT), leaf area index (LAI), solar radiation, air temperature, and soil moisture.

Dry and wet deposition is implemented in the model, where wet deposition is included in the treatment of clouds. The dry deposition velocity is modeled by a multiple path resistance scheme according to Zhang et al. (2003) , where aerodynamic resistance and quasi laminar sublayer resistance are a function of friction velocity (Wesely et al., 2002). In addition, different contributions to the overall canopy resistance depend on photosynthetically active radiation, air temperature, water-vapor deficit, LAI, and friction velocity.

The EURAD-IM system includes 4 dimensional variational data assimilation (*4Dvar*) for initial state and emission rate optimization (Elbern et al., 2007). This assimilation algorithm comprises an adjoint model which integrates a signal backward in time. Being implemented into the modeling system, the adjoint model can be modified to quantify the history of selected airmasses according to Vogel et al. (2020). By switching of chemical conversions, the retroplume operator allows the identification and investigation of source regions of air parcels including the convolution and mixing of different airmasses.

## 3.3 Model Inputs

In the following, important model configurations and their realizations selected in this study are briefly described. As summarized in Table 1, two representative realizations – a reference and an alternative option – are selected for each model configuration. Most model configurations apply to the meteorological forecasts of WRF and transfer to the atmospheric chemical forecasts via its dependencies on meteorological conditions. Firstly, initial- and boundary conditions for WRF are provided



**Table 1.** Selection and description of considered model input sources. PX = Pleim-Xiu surface layer parameterization, Du = Dudhia short-wave radiation parameterization. For further information on parameterization schemes in WRF see e.g Skamarock et al. (2008). Increased roughness lengths are based on Berndt (2018).

| input source | selected options | | description |
| --- | --- | --- | --- |
| | reference | alternative | |
| global meteo | IFS | GFS | global meteorological forecasts for initial- and boundary conditions in WRF |
| land use | USGS | MODIS | land use information (spatial distribution of PFT, LAI) in WRF and EURAD-IM |
| land surface | Pleim-Xiu | RUC | land surface model in WRF |
| boundary layer | MYJ + Eta | ACM2 + PX | boundary layer- and surface layer parameterization schemes in WRF |
| microphysics | WSM6 | TGS | cloud microphysics parameterization scheme in WRF |
| radiation | RRTMG | Du + RRTM | short- and longwave radiation parameterization schemes in WRF |
| roughness length | original | increased | vegetation dependent roughness length (Z0) in WRF |
| drought response | linear | no SMOIS | decrease of biogenic emissions for low soil moisture (SMOIS) |

by global meteorological analyses. Here the IFS reanalysis (Hortal, 1998) provided by ECMWF is used as reference and the operational GFS analysis from NOAA (Caplan et al., 1997) serves as an alternative option.

Secondly, reference information on land surface and vegetation types are given in the form of USGS (*U.S. Geological Survey*) land use categories. In WRF, information on surface types are provided by the GLCC (*Global Land Cover Characteristics*) database. Based on AVHRR (*Advanced Very High Resolution Radiometer*) observations between April 1992 and March 1993, 205 the surface at each location is classified as a single USGS land use category (Loveland et al., 2000). The USGS data base includes 24 different categories including water, urban, snow and ice as well as various vegetated surface categories (Anderson et al., 1976). Although the database provides unsupervised surface classification, the occurrence of different surface types is treated by mixed categories (e.g. "Cropland/Woodland Mozaic").

Land use information based on MODIS (*MODerate-resolution Imaging Spectroradiometer*) observations are selected as 210 alternative input. Vegetation products from MODIS and Sentinel-2 satellites can give more recent information on spatial distributions and also temporal evolution of vegetation types. Currently, Sentinel-2 provides vegetation products in the highest resolution (up to 10 m horizontal resolution, e.g., Immitzer et al., 2016; Drusch et al., 2012). However, these data are not available for 2012 as the satellite was launched in 2015. Thus, MODIS fractional vegetation data with 1 km spatial resolution (Friedl et al., 2002) are transferred to land use information for this study. Multiple studies indicate a more detailed and reliable 215 characterization compared to AVHRR based products (e.g., Smirnova et al., 2016; Hansen et al., 2002). However, transferring MODIS data to land use categories requires additional information on urban areas, water, snow and ice. Assuming an appropriate representation of these basic surface types, the related information of USGS were also used in the MODIS based classification. If the MODIS categories do not sum up to 100 %, the missing fraction is defined according to the USGS land use categories, if they are non zero.





Thirdly, WRF offers various options for different kinds of parameterizations. For this study, sensitivities to the land surface model (LSM), boundary- and surface layer parameterizations and cloud microphysics parameterizations as well as short- and longwave radiation parameterizations are considered. The selection of parameterization schemes is based on the frequency of usage (according to $https://www2.mmm.ucar.edu/wrf/users/wrf\_physics\_survey.pdf$, last access on 06.04.2020) and differences in the underlying approach. For example, the two layer Pleim-Xiu (Pleim and Xiu, 1995; Xiu and Pleim,

2001) and the multi layer RUC (*Rapid Update Cycle,* Benjamin et al., 2004) schemes serve as reference and alternative option for LSM. The MYJ (*Mellor-Yamada-Janjic,* Janjic, 1994) and ACM2 (*Asymmetric Convection Model version 2,* Pleim, 2007) boundary layer parameterizations are selected because of their $1.5^{th}$ order local- and $1^{st}$ order non local approach, respectively. As boundary- and surface layer parameterizations are closely related, the Eta-similarity (Janjic, 1996) and Pleim-Xiu (Pleim, 2006) surface layer parameterizations are selected with respect to the boundary layer parameterizations and resulting effects

are investigated together. For cloud microphysics, the WSM6 (*WRF Single-Moment 6-class,* Hong and Lim, 2006) and TGS (*Thompson Graupel Scheme,* Thompson et al., 2008) parameterizations are selected. The RRTMG (*Rapid Radiative Transfer Model for GCMs,* Iacono et al., 2008) scheme is used as reference for short- and longwave radiation while alternative options are Dudhia (Dudhia, 1989) and RRTM (*Rapid Radiative Transfer Model,* Mlawer et al., 1997), respectively. No effect of cumulus parameterization schemes is investigated here because the parameterization is switched off as the 1 km domain is

assumed to resolve related processes.

    Finally, one additional model configuration is considered for pollutant transport and dry deposition as well as biogenic emissions, respectively. The effect of drought response to biogenic emissions introduces an important source of uncertainty. No response to soil moisture is implemented in the MEGAN 2.1 code ($http://lar.wsu.edu/megan/docs/meganv2.10\_beta.tar.gz$, last access on 09 July 2018), but Guenther et al. (2012) propose a linear reduction of isoprene emissions below a threshold soil

moisture. While a general reduction of biogenic emissions under dry conditions is indicated by multiple studies, the explicit dependency of emitted gases is still under discussion (e.g., Pegoraro et al., 2004; Lavoir et al., 2009; Wu et al., 2015). In this study, the linear decrease of emissions proposed by Guenther et al. (2012) is implemented as reference for all biogenic gases while no dependency is assumed as alternative option ("no SMOIS"). Regarding transport, Berndt (2018) indicate that predefined values of roughness length in WRF underestimate true roughness in Europe. Thus for pollutant transport and dry

deposition, effects of increased vegetation dependent roughness lengths are investigated based to the values used by Berndt (2018).

## 4   Effects on Model Processes

Effects of different types of model inputs and configurations are investigated for biogenic emissions (Sect. 4.1), dry deposition velocities (Sect. 4.2) and pollutant transport (Sect. 4.3). This section focuses on the results for 12 July 2012, which are discussed

in more detail. All three model processes discussed here are highly dependent on meteorological fields. However, using an existing meteorological ensemble appears to be not sufficient for this application. An investigation of sensitivities from the



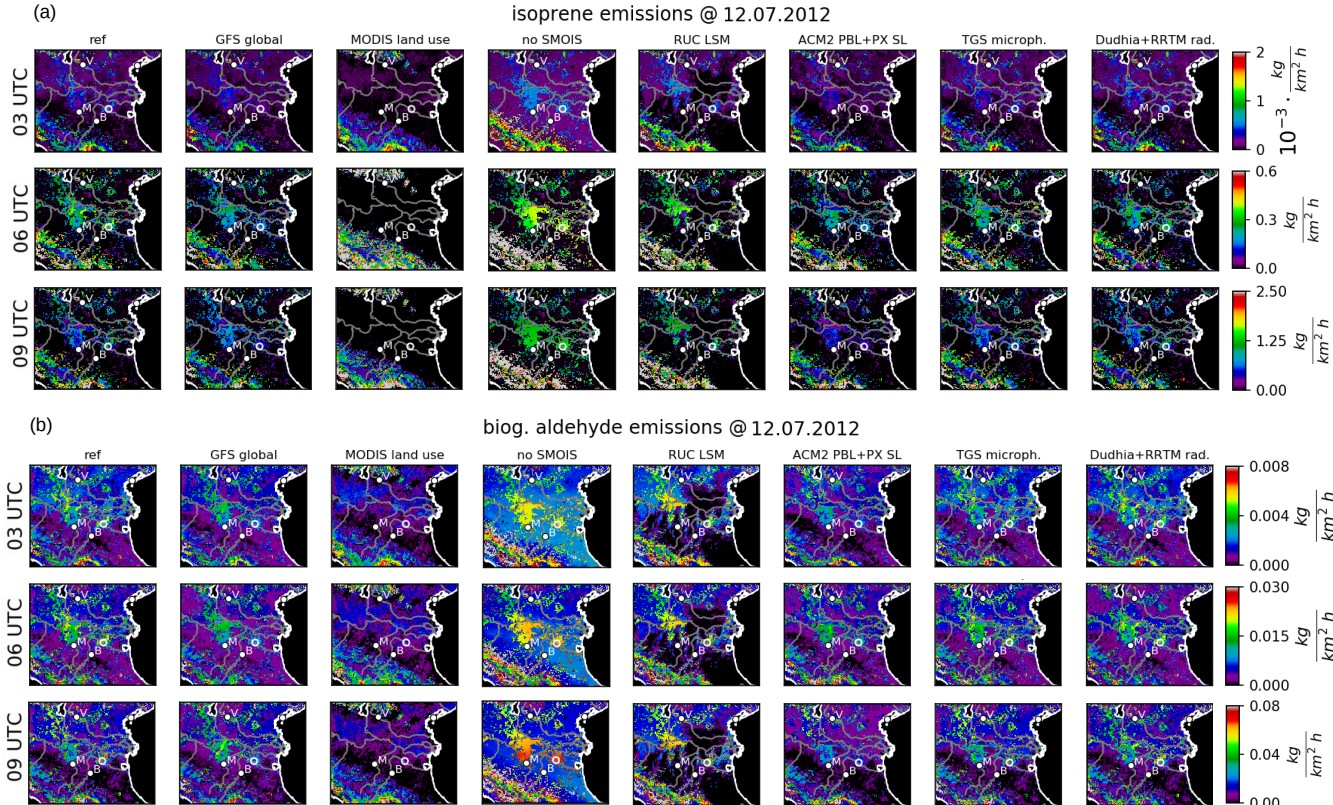

**Figure 2.** Isoprene (a) and biogenic aldehyde (b) emissions on 12 July 2012 at 03, 06 and 09 UTC (coded by colors) for different model configurations: reference, GFS global meteorology, MODIS land use, and no response to soil dryness ("no SMOIS"), RUC LSM, ACM2 boundary layer + Pleim-Xiu surface layer schemes, TGS microphysics and Dudhia shortwave- + RRTM longwave radiation. Some important cities (Verona, Bologna, Modena) are indicated by their initial letters. The location of the Zeppelin observations on this day is given as small circle.

global GFS (*Global Forecast System*) ensemble from NOAA did not induce significant differences in the simulated boundary layer in this case (not shown).

## 4.1 Effects on Biogenic Emissions

The effects of different model configurations on biogenic emissions of isoprene and aldehyde are given in Fig. 2. Note that biogenic aldehyde emissions from MEGAN 2.1 include total emissions from acetaldehyde and a set of higher aldehydes which are not treated individually (compare Guenther et al., 2012).

In general, biogenic emissions increase significantly after sunrise due to increasing solar radiation. Differences between nighttime (03 UTC) and daytime (09 UTC) emissions are more significant for isoprene than for aldehyde. This is because

isoprene is a direct product of photosynthesis which is mainly limited to daytime conditions. For the reference setup (Fig. 2,





"ref"), daytime isoprene emissions are mainly restricted to the Apennine Mountains and two areas within the the central Po valley north of Modena and Bologna. According to USGS land use, these locations are assigned to "Deciduous Broadleaf Forest" and "Crop/Woodland Mosaic", respectively. In contrast to "Dryland Cropland and Pasture" in the rest of the valley, broadleaf trees emit high levels of isoprene. Thus, even small numbers of trees result in significantly increased local isoprene emissions.

In these regions, increased biogenc emissions are also found for aldehyde. However, the differences between different land use types remain small compared to isoprene.

The high dependency on tree coverage is emphasized by comparing reference biogenic emissions to emissions based on MODIS land use (Fig. 2, "land use"). In contrast to USGS, MODIS does not indicate any trees within the Po valley, which results in neglectable isoprene emissions in this region. At the same time, the whole Apennine Mountains and southern foothills

of the Alps are assigned to high coverage of broadleaf trees resulting in high biogenic emissions. The use of GFS global meteorology does not change the general emission patterns (Fig. 2, "global"). Caused by different initial- and boundary conditions, slight differences are found in different part of the domain for both gases. The implemented response of biogenic emissions to soil dryness significantly influences biogenic emissions (Fig. 2, "no SMOIS"). By neglecting this response, emissions are considerably larger than for the reference case, especially in the southern part of the domain. As soil moisture decreases after

sunrise, the largest sensitivities are found at 09 UTC for both gases.

The RUC LSM induces slightly increased biogenic emissions of both, isoprene and aldehyde, in most areas (Fig. 2, "LSM"). In contrast, emissions are reduced to almost zero in the south-eastern parts of the Po valley. This reduction is caused by low soil moisture predicted by RUC LSM in the morning hours which results in drought induced plant stress. The combined effect of boundary layer- (PBL) and surface layer (SL) schemes is found to be small for both gases (Fig. 2, "PBL + SL"). Reduced

biogenic emissions are predicted by ACM2 PBL + Pleim-Xiu SL compared to the reference using MYJ PBL + Eta SL schemes. While these differences in isoprene are mainly restricted to areas of high emissions in the central Po valley, the reduction is more extended for aldehydes. Only minor changes in biogenic emissions due to microphysics- and radiation schemes are visible (Fig. 2, "microph.", "rad."). Using TGS microphysics instead of the reference WSM6 does only induce small local effects during nighttime (03 UTC). Although being small, effects of using different radiation schemes after sunrise can be

attributed to different formulations of shortwave radiation by the Dudhia and RRTMG schemes.

### 4.2 Effects on Dry Deposition Velocities

Dry deposition velocities are only investigated for aldehydes as isoprene is not directly dry deposited in the model. The dry deposition velocities of aldehydes in Fig. 3 differ significantly between daytime and nighttime conditions. While large deposition velocities are restricted to the peaks of the Apennines at 03 UTC, spatial differences reduce after sunrise. In the

central Po valley, aldehyde deposition velocities at 03 UTC are approximately one order of magnitude smaller compared to 06 and 09 UTC. A small overall reduction of deposition velocities is simulated for GFS global meteorology compared to the ECMWF reference (Fig. 3, "global"). Large differences in deposition velocities are found with respect to land use (Fig. 3, "land use"). Most prominent is a reduction in urban areas according to USGS land use information. This is caused by an imperfect overlap of non vegetated regions in MODIS and urban land use in USGS. In the rest of the Po valley, effects on dry deposition





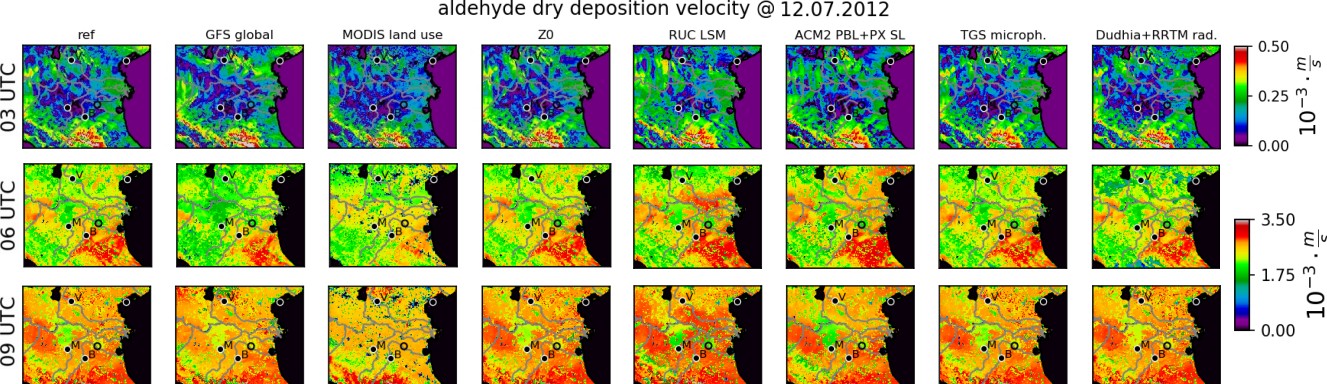

**Figure 3.** Dry deposition velocities of aldehyde on 12 July 2012 at 03, 06 and 09 UTC (coded by colors) for different model configurations including increased roughness length ("Z0"). Plotting conventions as in Fig. 2.

velocities point into different directions in different regions. While dry deposition is decreased in the central valley, increased values are found in the southern Apennines at 09 UTC. Increasing roughness length does slightly increase dry deposition velocities after sunrise (Fig. 3, "Z0").

Using the RUC LSM results in larger dry deposition velocities in the central Po valley at all times. Boundary layer- and surface layer schemes induces local effects with opposite signs after sunrise south of Modena and east of Verona, respectively. The selection of the microphysics schemes induces only minor local changes in this case (Fig. 3, "microph."). The largest effect of Dudhia shortwave and RRTM longwave radiation schemes appears at 06 UTC, were dry deposition velocities are decreased in the northern part of the domain (Fig. 3, "rad.").

## 4.3 Effects on Pollutant Transport

This section investigates effects on transport of any atmospheric pollutant. Section 4.3.1 evaluates effects on fields of friction velocity forecasted by WRF-ARW. As friction velocities does only provide information on the horizontal wind speeds determining pollutant advection, source regions of airmasses indicate total effects of atmospheric dynamics. An analysis of source regions of an exemplary airmass is given in Sect. 4.3.2.

### 4.3.1 Effects on Friction Velocities

In general, friction velocities does not change substantially between night- and daytime but increase in most areas with increasing local instability. For the reference setup in Fig. 4, high values predicted at 03 UTC over the peaks of the Apennines are almost one order of magnitude higher than in the rest of the domain. After sunrise, these large differences reduce over time by increasing mean values and slightly decreasing peak values. GFS global meteorology indicates reduced friction velocities in the central Po valley especially at 06 UTC (Fig. 4, "global"). Effects of land use information appear to influence friction velocities on comparably small scales (Fig. 4, "land use"). Using MODIS instead of USGS, friction velocities are partly increased in the




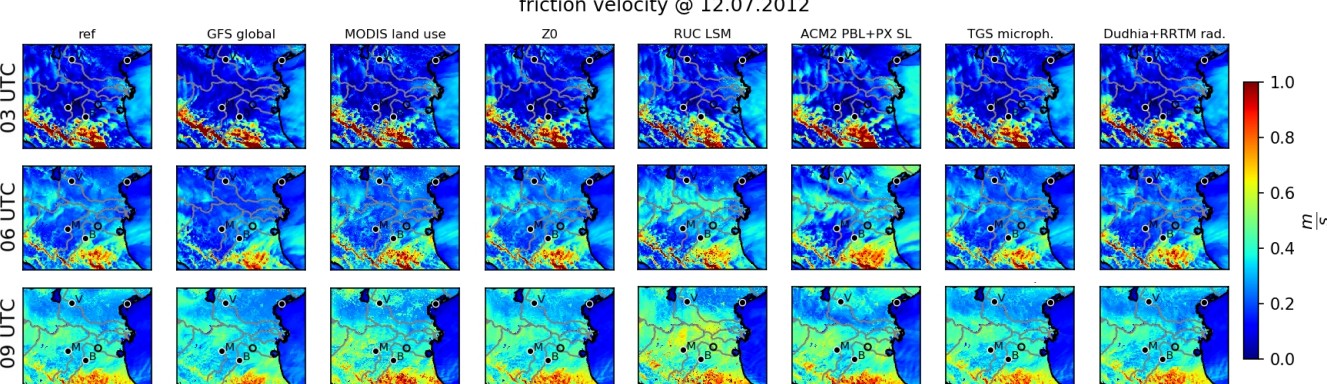

**Figure 4.** Friction velocities on 12 July 2012 at 03, 06 and 09 UTC (coded by colors) for different model configurations including increased roughness length ("Z0"). Plotting conventions as in Fig. 2.

north-eastern part of the domain. Friction velocities appear also to be little sensitive to increased roughness length, inducing slightly increased values after sunrise (Fig. 4, "Z0").

The selection of the LSM as well as boundary layer- and surface layer schemes induce significant effects on friction velocities (Fig. 4, "LSM"+"PBL+SL"). As early as 03 UTC, the RUC LSM triggers a region of increased values south of Lake Garda. This signal penetrates along the Po river affecting large parts of the valley from 06 to 09 UTC, resulting in almost doubled friction velocities. Similar to the sensitivity to the LSM, friction velocities are partly increased in the central Po valley by the ACM2 boundary layer- and Pleim-Xiu surface layer schemes. However, this effect is less pronounced and regions of high values change with time. At the north-eastern edge north of Venice, friction velocities are increased for all times. Microphysics- and radiation parameterizations only induce minor changes in friction velocities (Fig. 4, "microph.","rad."). Slightly changed patterns due to Dudhia shortwave and RRTM longwave radiation schemes are observed in the western and eastern Po valley at 06 and 09 UTC, respectively.

### 4.3.2 Effects on Source Regions

Some studies are available that investigate the history of airmasses in the Po valley by the means of long term backward trajectories (Sogacheva et al., 2007; Pernigotti et al., 2012; Sullivan et al., 2016). However, this approach does not account for local effects and related uncertainties in transport and mixing. Here, these effects are analyzed using retroplume calculations for an exemplary airparcel. The selected airmass is located at the position of the Zeppelin observations ($44.7°N$, $11.6°E$, "*target location*") on 12 July 2012 at 06 UTC ("*target time*") in 100 m height above sea level. Starting at this time, the retroplume calculation provides relative contributions of source areas of this airmass backward in time. The Zeppelin observations were perfromed close to the base station of the Zeppelin at San Pietro Capofiume (SPC) in the south-eastern part of the Po valley. Being located approximately 30 km northeast of Bologna, SPC is classified as a urban background site (Kaiser et al., 2015;





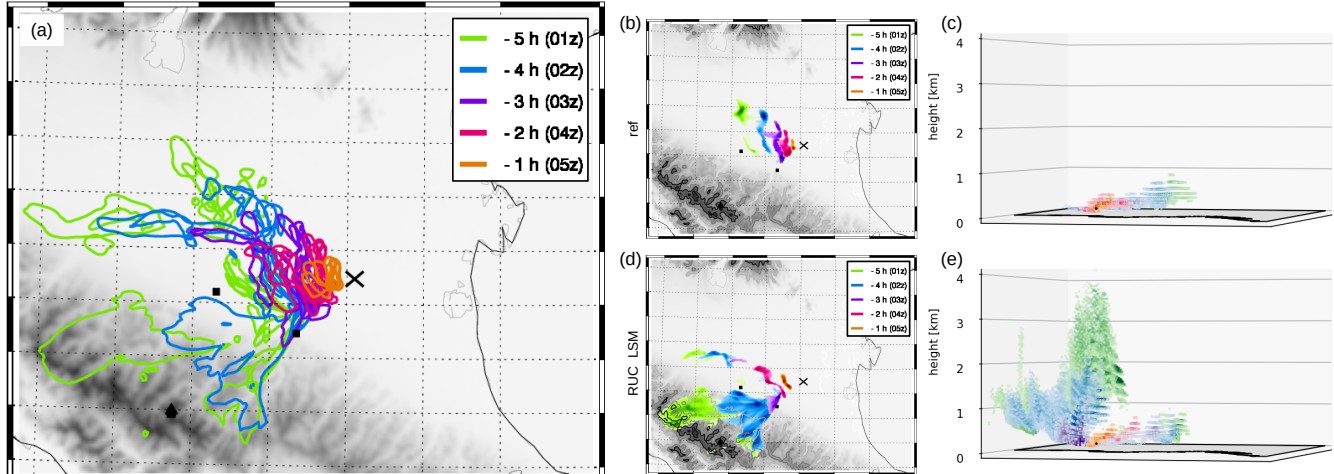

**Figure 5.** Horizontal ( (a),(b) and (d) ) and vertical ( (c) and (e) ) distributions of source regions for $44.7° N$, $11.6° E$ and 100 m a.s.l. (black cross, "target location") at 12 July 2012, 06 UTC. (a): Significant contributions to vertically integrated source regions for all model configurations (incl. increased roughness length) are given as isolines (colored by time according to legend). Gray colors indicate the surface topography. (b) and (c): reference configuration, (d) and (e): RUC LSM. For (b) and (d), the viewing direction is from east-northeast towards west-southwest.

Rosati et al., 2016b). However, Sandrini et al. (2016) state that it might be affected by the Bologna urban area in case of south-westerly winds.

    Figure 5(b)+(c) show the evolution of source regions for the reference setup. The selected airmass is advected from western to north-western directions due to slow westerly winds. During the last 3 hours before the target time, the airmass converged horizontally by meredional mixing processes. Thus, contributions of this airmass converge from south-western to north-western

directions at this time. 5 hours before, the major source of the airmass is found north-west of the target location. During the entire time interval, the vertical extension of source areas remains below 1 km altitude (Fig. 5(c) ). This is caused by low vertical mixing, which is typical for the early morning hours over flat terrain.

    By applying different kinds of model configurations, resulting effects on horizontal source regions are shown in Fig. 5(a). Horizontal source areas start to diverge already during the first hours before the target time. Three hours earlier, significant

contributions span from Bologna in the south-west for RUC LSM to the western central Po valley in the north-west for GFS global meteorology. Five hours before, additional differences in transport distance and vertical mixing become visible. Transport distances from the selected target location varies more than a factor of two at this time. For example, source regions for the ACM2 boundary layer- and Pleim-Xiu surface layer schemes almost extend to the western boundary of the domain within 5 hours. This is caused by increased turbulent transport compared to the reference MYJ boundary layer- and Eta surface layer

schemes. In contrast, Dudhia and RRTM radiation schemes as well as MODIS land use information indicate slow horizontal



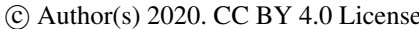

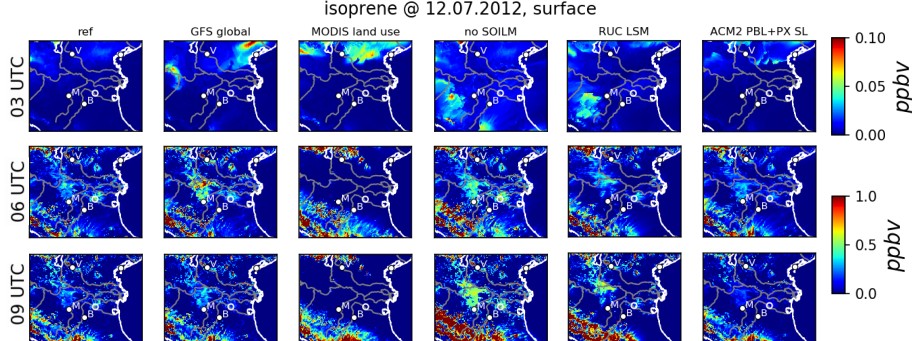

**Figure 6.** Isoprene concentrations on 12 July 2012 at 03, 06 and 09 UTC (coded by colors) for different model configurations. Plotting conventions as in Fig. 2.

transport from urban areas close to Modena and Bologna. Compared to mainly agricultural source areas for the reference and ACM2, these slight changes in local dynamics may result in significant changes in the composition of this airmass.

As shown in Fig. 5(d)+(e), the RUC LSM has the largest effect on the simulated airmass history. In the horizontal plain, a major contribution to the airmass was transported from south-western directions crossing Bologna. Prior ti this, it descended

from the Apennine Mountains where high winds speeds advect air from comparably remote source regions. This is related to increased mixing processes which result in large extended source regions 4 hours before. Between 4 and 5 hours before, the source areas shows convergence of two distinct airmasses flowing around the highest peaks of Monte Cimone with altitudes of more than 2000 m a.s.l. (Fig. 5(d) ). The overflow above the Apennines forces the airmass to source altitudes of up to 4 km a.s.l. 5 hours before the target time. Additionally, a small contribution of the airmass originates from a narrow valley south-west

of Bologna with lower wind speeds and source altitudes of below 1 km a.s.l. (lower left part of Fig. 5(e) ).

## 5 Effects on Surface Concentrations

Simulations of surface near concentrations of trace gases are controlled by the model processes as discussed above. Resulting effects of model configurations on surface concentrations are presented for isoprene as short lived, directly emitted BVOC and aldehydes as set of oxidized BVOC. As microphysics and radiation schemes appear to hove only minor effects on the model

processes in this case, their effects on concentrations is not further discussed.

### 5.1 Effects on Isoprene

Being a short lived biogenic trace gas, isoprene concentrations are mainly determined by changes in biogenic emissions as shown in Fig. 6. Consequently, effects of land use information and drought response on isoprene emissions transfer almost directly to differences in surface concentrations. Using MODIS land use produces negligible isoprene concentrations within





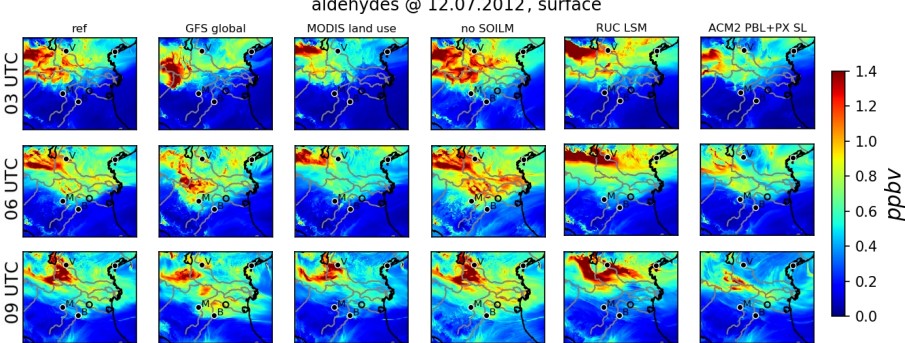

**Figure 7.** Aldehyde concentrations on 12 July 2012 at 03, 06 and 09 UTC (coded by colors) for different model configurations. Plotting conventions as in Fig. 2.

the Po valley and increased concentrations in the mountains. An overall increase in isoprene concentrations is achieved at all times when no emission response to soil dryness is considered.

Nevertheless, effects of atmospheric transport does also influence isoprene concentrations when changing modeled global meteorology as well as LSM and boundary layer schemes. At 06 UTC, surface concentrations north of Modena are increased for GFS global meteorology although local emissions were slightly reduced (compare Sect. 4.1). This is caused by higher

atmospheric stability leading to less dispersion which is indicated by reduced friction velocities (compare Sect. 4.3.1). For the RUC LSM, increased isoprene emissions are partly compensated by increased mixing in the central valley after sunrise. Thus, the most significant increase in surface concentrations is found in the Apennine Mountains. Effects of combined boundary layer- and surface layer schemes do appear on local scales due to slightly reduced emissions and variable dynamical effects.

## 5.2   Effects on Aldehydes

Due to long lifetimes compared to isoprene, surface concentrations of aldehydes in Fig. 7 are spatially smooth with highest values in the north-western model domain. Combined effects of changes in biogenic emissions, dry deposition and pollutant transport may affect the single components differently. As drought response does only affect biogenic emissions by definition, increased surface concentrations in the western valley are solely caused by higher biogenic aldehyde emissions when the response is neglected. Similarly, decreased aldehyde concentrations in the central valley for MODIS land use are induced by

lower biogenic emissions rather than higher dry deposition (compare Sect. 4.1).

Changes in the Po valley by global meteorology, the LSM and boundary layer schemes are mainly attributed to changed flow patterns, leading to increased aldehyde concentrations in this region. This is caused by north-westerly flow transporting aldehydes from areas of locally increased biogenic emissions north of Modena (compare Sect. 4.3.2). In contrast, south-westerly flow is induced by the RUC LSM advecting clean well mixed air from the Apennines. Thus, small differences in boundary

layer dynamics lead to variations in local aldehyde concentrations of more than a factor of two in this case. Additionally, high

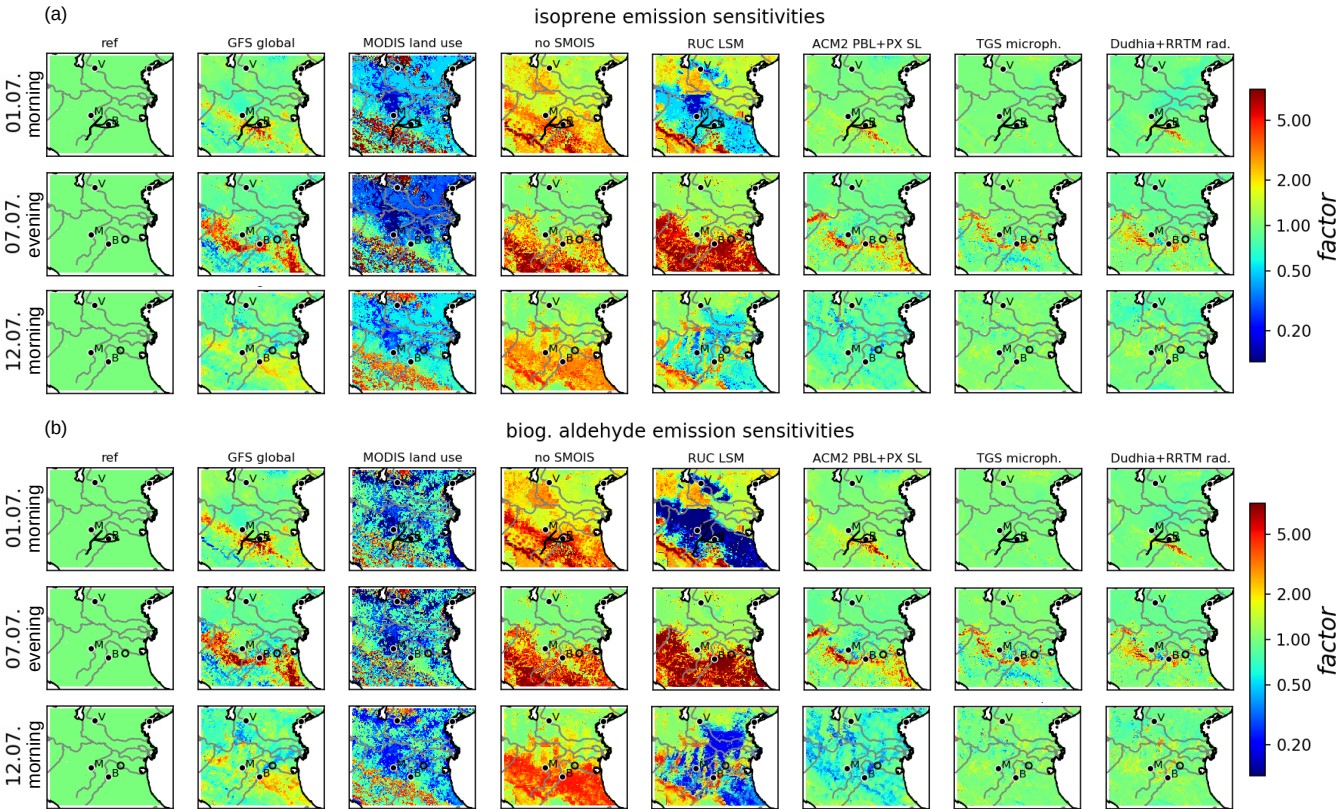

**Figure 8.** Averaged sensitivities on isoprene (a) and biogenic aldehyde (b) emissions on 01 July morning (00–10 UTC), 07 July evening (12–22 UTC), 12 July 2012 morning (00–10 UTC) (temporally averaged factors w.r.t. reference, coded by colors) for different model configurations. Plotting conventions as in Fig. 2.

concentrations in the north-west and southern Apennines for RUC LSM are caused by increased biogenic emissions in this area.

## 6   Generalized Effects of Biogenic Emissions

The case analysis in Sect. 4.1 indicates almost constant effects of model configurations, especially for biogenic emissions. Thus,
the results can be comprised to temporally averaged sensitivities on biogenic emissions for a set of representative cases during the campaign. Sensitivity factors are calculated as temporal average of emissions at each time divided by their corresponding values from the reference configuration. While for 01 July and 12 July 2012, the analysis focuses on the morning hours between 00 and 10 UTC, evening conditions between 12 and 22 UTC are investigated for the 07 July 2012 case. Note that minimum emissions of $1.0 \cdot 10^{-3} \frac{kg}{km^2 \, h}$ are defined in order to avoid unrealistic large factors induced by low values.




As shown in Fig. 8, sensitivities of both biogenic gases during the campaign are most significant for land use, drought response and the LSM. As expected, the large averaged sensitivities to land use information are highly similar for all cases and both gases. Somehow lower averaged sensitivities on isoprene on 01 July and 12 July are induced by emissions below the minimum value in the early morning. A strong increase of emissions due to neglected drought response is found for all cases, which most significant effects in the southern part of the domain.

For the two morning cases (01 July , 12 July ) RUC LSM results in reduced emissions in most parts of the Po valley and increased values south of Verona and in the south-west. In contrast, highly increased biogenic emissions are produced in the entire southern half of the domain during the evening case (07 July ). These diurnal differences are caused by general different temporal evolutions of soil moisture, which limit biogenic emissions. While soil moisture from Pleim-Xiu LSM drops during daytime and recovers fastly during nighttime, RUC predicts a continuous decrease during these days. Thus, lower soil moisture
for RUC induces lower emissions in the morning, and vise versa in the evening. Despite this general diurnal differences, sensitivities to the LSM appear to be mostly constant during the campaign.

The use of GFS global meteorology induces small, yet similar effects for all cases. While emissions are increased in the northern Apennines, emissions tend to be slightly lower elsewhere. Effects of boundary layer- and surface layer schemes remain small and differ between the cases. While for 12 July the ACM2 boundary layer- and Pleim-Xiu surface layer schemes
induce decreased biogenic emissions, they produce increased emissions north of the Apennines in the other cases. Microphysics and radiation schemes induce only significant effects in the evening case (07 July ) with slightly increase emissions north of the Apennines.

## 7   Conclusions

Forecasts of biogenic trace gases in the PBL are well known to be subject so large uncertainties. The PEGASOS campaign in the
Po valley 2012 provides a challenging, yet common environment for investigating model uncertainties with focus on biogenic gases in the PBL. As a precursory step to probabilistic model evaluation with PEGASOS observations, this study identifies and quantifies principal sources of forecasts uncertainties induced by model configurations via different model processes. Two major pathways of model uncertainties with respect to model parameters and -inputs are identified in this study:

Firstly, exceptionally large differences in simulated biogenic emissions are found in the Po valley during the campaign.
Dominating impacts of land use information, the land surface model and emission response to drought state a high sensitivity of biogenic emissions to various land surface conditions. Although biogenic emissions are known to be affected by environmental conditions, these three effects induce unexpectedly high variations of up to an order of magnitude in the studied case. Due to the common approach used for modeling biogenic emissions, the results appear to be highly correlated for different biogenic gases. Moreover, these effects are found to be almost constant during the campaign, varying only with the diurnal cycle. Despite
invariant vegetation distributions, this persistence is forced by steady meteorological conditions during the campaign.

Secondly, the model configuration also highly influences regional flow patterns with effects on pollutant transport and mixing. Especially the selection of land surface model and boundary layer schemes induce significant differences in local pollutant





transport. Although effects on friction velocities remain small, a retroplume analysis proves high sensitivities of pollutant transport to the selected model configurations. Most prominent changes in horizontal advection and vertical mixing patterns
induced by the land surface model can be attributed to differences in surface heat exchange. In this study, the selection of cloud microphysics and radiation schemes does only induce minor uncertainties during this study. This is due to the calm and sunny conditions in the Po valley and may be more significant for other meteorological conditions.

As a result, surface concentrations of biogenic trace gases show large sensitivities to model configuration. Differences in dry deposition velocities does not contribute significantly to changes in surface concentrations of all gases. Sensitivities of
isoprene concentrations are mainly induced by changes in land use information, the land surface model and drought response via their impact on emissions. Surface concentrations of aldehydes appear to be highly sensitive to those model configurations, which affect biogenic emissions and pollutant transport. Where the sensitivities to each model configuration varies in space and time depending on the contribution of local biogenic- and anthropogenic emissions to total concentrations. Areas with high contributions of biogenic emissions are highly sensitive to land surface conditions whereas areas with low total emissions are
more affected by flow patterns. For example, although San Pietro Capofiume (SPC) is surrounded by agricultural landscapes, it appears to be potentially affected by urban areas depending on simulated boundary layer transport. Moreover, boundary layer transport is found to vary significantly with the model configuration, most notably with respect to the land surface model and boundary layer scheme. Thus, especially in areas with small scale emission patterns, the selection of the model configuration can result in highly variable local concentrations. This effect was corroborated by diverging source regions of an exemplary
airmass and is thus likely to apply also to non biogenic gases.

Although the presented results are specific for the conditions in Po valley, it can be claimed that forecasts of biogenic gases would generally benefit substantially from improved representation of the land surface. While improved land use information could be retrieved from more recent satellite products like Sentinel-2, considerably different effects of land surface models for both, biogenic emissions and pollutant transport underline the significance of soil moisture estimates for air quality modeling.
Furthermore, the large amount and complexity of sensitivities found in this study demonstrate the need to account for forecast uncertainties with special focus on biogenic emissions and pollutant transport. This is especially important for model evaluation with observations as well as chemical data assimilation in the PBL. A follow up study will evaluate probabilistic model forecasts of biogenic gases using airborne PEGASOS observations of biogenic gases. In this context, the sensitivities derived in this study provide a basis for estimating forecast uncertainties with respect to different model processes.

*Data availability.* The model data produced for this study is stored locally at the Rhenish Institute for Environmental Research as well as at the Jülich Supercomputer Centre (JSC) of Research Centre Jülich. It is available by request via email (av@eurad.uni-koeln.de).

*Author contributions.* AV performed the simulations, analyzed the results and wrote the manuscript. HE supervised the work, contributed significantly to the analysis and helped in the preparation of the manuscript.



*Competing interests.* The authors declare that they have to competing interests.

*Acknowledgements.* This work has been funded by the Helmholtz Climate Initiative REKLIM (Regional Climate Change), a joint research project of the Helmholtz Association of German research centers (HGF) under grant: REKLIM-2009-07-16. The author gratefully acknowledges the computing time granted through JARA-HPC on the supercomputer JURECA (Jülich Supercomputing Centre, 2018) at Forschungszentrum Jülich. This work would not have been possible without the meteorological analysis obtained from the NCEP's Global Forecasting System (GFS).



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
