# Peer review of "Identifying forecast uncertainties for biogenic gases in the Po valley related to model configuration in EURAD-IM during PEGASOS 2012"

_Atmospheric Chemistry and Physics, 2020_

## Referee Comment (RC1) · Anonymous Referee #1 · 24 Nov 2020

This paper presents a very interesting study analyzing a variety of meteorological and physical impacts on biogenic emissions and the resulting ambient concentrations. Gaining a greater understanding of these processes, their impacts and uncertainties is of great importance for air quality modeling. The design of the experiment, changing a variety of model parameterizations in turn, provides a wealth of data for this study. The analysis of the impacts on biogenic emissions (isoprene and lumped aldehydes) is interesting and useful for the community.

However, the results on the effect on surface concentrations are presented without any

consideration of the atmospheric chemistry that might affect their results. Isoprene reacts with OH very quickly, and OH distributions are likely influenced by the meteorological changes (clouds, humidity). This should at least be mentioned and preferably OH fields also shown, to allow a greater understanding of the changes in isoprene among cases. It would also be interesting to use the sensitivity studies performed for this work to analyze the intensity of segregation and see how the model parameterizations affect that and thus the chemistry. See Kaser et al., GRL, 2015, doi:10.1002/2015GL066641.

I have greater concern about the aldehydes results because aldehydes have a large secondary production that is not even mentioned in the paper. So the surface concentrations will be affected not only by biogenic emissions but chemical production (from isoprene and anthropogenic sources). It might have been better to study something like methanol, which has large biogenic emissions, longer lifetime and no secondary production. Thus, in order to keep Section 5 in the paper, much more discussion should be included about the impact of atmospheric chemistry on the surface concentrations.

Some additional comments:

l.90: aldehydes are also photochemically produced.

l.236+: You might want to refer to Jiang et al., https://doi.org/10.1016/j.atmosenv.2018.01.026 for a discussion of the implementation of drought impact in MEGANv3.

Figure 4: It is very difficult to pick out the differences highlighted in the discussion in these tiny panels. It would be nice to find another way to illustrate these differences. Perhaps all except the first column should be differences (percent) from the ref case. Or show just averages over the Po Valley and other regions of particular relevance (not on a map).

Figures 6 & 7 are also too small - they could at least be enlarged to the width of the page, but difference plots would help illustrate features.

There are a number of spelling and grammar errors, but the paper is understandable. Here are some corrections:

l.112: through -> trough

l.269: 'neglectable' -> negligible

l.305: 'friction velocities does only' -> 'friction velocities only'

l.307: do you mean 'exemplary' (best of its kind) here, or perhaps just 'example' or 'representative'? [similarly elsewhere in the paper]

l.309: does -> do

l.353: plain -> plane

l.364: hove -> have

l.419 'so large' -> 'to large'

l.420 add comma after 'common'

---

## Referee Comment (RC2) · Anonymous Referee #2 · 21 Dec 2020

This paper presents the uncertainties associated with the simulation of some of the biogenic gases depending on the meteorological model settings and land-use datasets. The presented simulation scenarios are focused on the PEGASOS field campaign conducted over the Po valley during summer 2012. Three flight cases are selected for the numerical experiments.

There are large uncertainties in simulating the fluxes of the biogenic volatile organic compounds and their mixing ratios in the air quality models. It's worth studying these uncertainties. However, my primary concern here is that the authors run the WRF

model with different input data and physics schemes without verifying how suitable are the selected model settings for the given task. Here, the role of the WRF model is to provide the meteorological drivers to the EURAD-IM offline chemistry transport model. The authors don't provide any verification of the model simulations using the surface or aircraft observations. It's likely that some of the presented model scenarios aren't able to accurately simulate the meteorology in the Po valley for the selected days. Also, accurate simulation of the soil moisture in WRF usually requires model spin-up over some time period, so using a "good" land-surface scheme isn't sufficient. Thus, using inaccurate or unverified meteorological simulations to drive the offline EURAD-IM chemistry model doesn't make sense.

One of the selected sensitivity simulations is done using the MODIS versus USGS land-use dataset. It's expected that the differences in the vegetation map for the Po valley will lead to large differences in the fluxes of the biogenic VOCs. However, the MODIS data is more up-to-date than the USGS land-use data. Therefore, it isn't clear what we learn by testing both meteorological and air quality models using the old (probably not accurate) land-use data (USGS) as input.

The analysis of the effects of source regions (section 4.3.2) is interesting, but again without the verification of the wind speed/direction and other meteorological variables, it's impossible to determine which model simulations or scenarios are realistic here.

It's possible that some of the WRF model scenarios are somewhat similar in terms of their forecast skill, but at least basic model verification is required to select such model configurations to conduct reasonable meteorological simulations to be used as input in the chemistry transport model.

The simulation of other terpenes (e.g. alpha-pinene) isn't presented here. Instead, the authors present the simulation of the aldehydes, which are also produced by the gas chemistry in the EURAD-IM model. This aspect requires additional analysis. Again, it's hard to make any conclusions with respect to the accuracy of the gas chemistry simulations as none of the simulated chemical species are compared against the aircraft or other measurements. The discussion of the dry deposition is interesting, however, the role of more important processes such as photochemistry is necessary to consider.

Based on the aforementioned shortcomings of the study, I urge the authors to redo the numerical experiments, conduct extensive model verifications, and submit a substantially revised version of the paper in the future.

---

## Author Comment (AC1) · 25 Jan 2021

We thank the reviewer for the elucidating evaluation and valuable remarks. We did several substantial modifications to the manuscript as requested and we confident to address all remarks in a satisfying way.

**This paper presents a very interesting study analyzing a variety of meteorological and physical impacts on biogenic emissions and the resulting ambient con-**

[Figure]

**centrations. Gaining a greater understanding of these processes, their impacts and uncertainties is of great importance for air quality modeling. The design of the experiment, changing a variety of model parameterizations in turn, provides a wealth of data for this study. The analysis of the impacts on biogenic emissions (isoprene and lumped aldehydes) is interesting and useful for the community. However, the results on the effect on surface concentrations are presented without any consideration of the atmospheric chemistry that might affect their results. Isoprene reacts with OH very quickly, and OH distributions are likely influenced by the meteorological changes (clouds, humidity). This should at least be mentioned and preferably OH fields also shown, to allow a greater understanding of the changes in isoprene among cases.**

Reply1: We are grateful for this clue. In this context, OH provides useful insights in relation to isoprene concentrations and downstream reactive atmospheric chemistry. Therefore, we added an analysis of OH surface concentrations including Fig. 8 (Fig. 1 in this document) and Subsection 5.2 (l. 406-419 new count) in the manuscript (replacing aldehyde concentrations, as described below):

*" The hydroxy radical OH is a highly reactive oxidant in the atmosphere acting as most important sink of isoprene (Kaser et al., 2015). Generally, OH may be influenced by the model configuration via reaction with biogenically emitted gases and meteorological conditions. Local meteorology mainly affects OH by changes in radiation related to humidity and clouds. In this specific case, the weather in the Po region was continuously characterized by clear and dry conditions as described in Sect. 2.1. Thus, no significant differences in humidity and cloud coverage are simulated by the model configurations (not shown). This renders the differences in OH concentrations being determined by changed biogenic VOCs.*

*As expected from atmospheric chemistry, daytime OH concentrations shown in Fig. 8 are reduced in regions of high BVOC concentrations like the central Po valley and the southern Apennines. In contrast, OH concentrations remain comparably high in the mountains and over the ocean were isoprene concentrations are neglectable. This direct dependence of OH to*

*biogenic gases causes also significant differences in OH concentrations with respect to model configuration. In this case, the effects are most dominant in cases of increased isoprene concentrations with respect to the reference simulation. Significant reduction of OH is induced by excluding drought response ("no SMOIS"), RUC LSM, and less pronounced for MODIS land use in the southern Apennines. While these reductions are persistent in time, increased isoprene concentrations in the central Po valley for GFS global meteorology at 06 UTC result in temporally reduced OH concentrations in this region. "*

Related modifications have been made in the conclusions (l.467-470 new count) as well as the abstract (l.12-14 new count):

ABSTRACT: *" As a result, large sensitivities to model configuration are found for surface concentrations of isoprene as well as OH, affecting reactive atmospheric chemistry. "*

CONCLUSIONS: *" Moreover, changes in surface concentrations of biogenic trace gases induce significant differences in OH concentrations affecting reactive atmospheric chemistry. Excluding the emission response to drought stress reduces local OH concentrations by up to a factor of three in this study. "*

**It would also be interesting to use the sensitivity studies performed for this work to analyze the intensity of segregation and see how the model parameterizations affect that and thus the chemistry. See Kaser et al., GRL, 2015, doi:10.1002/2015GL066641**

Reply2: The investigation of dynamical separation of chemical compounds provides a very interesting aspect in the context of this study. We thank the reviewer for pointing towards this topic which is skillfully exposed in Kaser et al, 2015. As described above in Reply1, we added the evaluation of OH fields in the manuscript which provides useful information in this context. However, - in our study context and related objectives - we believe to perceive two critical points, which hinder an addition of segregation in the

current study:

1. In our current setup, the temporal resolution of emission information is not suitable for calculating fluctuations as required for the intensity of segregation. For computational reasons, MEGAN emissions are not calculated every timestep (currently: every 1800 sec.).

2. According to our opinion, a thorough investigation of segregation as proposed in Kaser 2015 would require a dedicated extra investigation of related processes which would be too voluminous to be added in this study.

Clearly, the Kaser et al. study clearly suggests such an investigation in a follow-up study. At this point we suggest to confine to advise the reader to this item and added a reference to Kaser 2015 as example for the complexity of chemistry-turbulence interactions in the introduction (l. 42-44, new count):

*" An example of highly complex chemistry-turbulence interactions is found by Kaser et al. (2015) who investigated effects of local separation of isoprene and OH. "*

**I have greater concern about the aldehydes results because aldehydes have a large secondary production that is not even mentioned in the paper. So the surface concentrations will be affected not only by biogenic emissions but chemical production (from isoprene and anthropogenic sources). It might have been better to study something like methanol, which has large biogenic emissions, longer lifetime and no secondary production. Thus, in order to keep Section 5 in the paper, much more discussion should be included about the impact of atmospheric chemistry on the surface concentrations.**

Reply3: We fully agree with the reviewer that aldehydes are affected by a large number of processes like secondary production. Individual effects of those cannot be separated by the approach used in this study. We therefore decided to withdraw the analysis of aldehyde concentrations from Section 5. Instead, we added the investigation of OH (as described above) and noted this issue in the beginning of Sect. 5 (l.386-389 new

count) as follows:

*" The evaluation of biogenic gas concentrations is restricted to isoprene because of its direct dependency on the model processes discussed above. Other biogenic gases are affected by additional processes like secondary production which hamper a detailed evaluation. Instead, resulting OH concentrations are analyzed on their impact on reactive atmospheric chemistry. "*

Unfortunately, studying methanol concentrations cannot be pertinently addressed in our modeling system. EURAD-IM uses the RACM-MIM chemistry mechanism which handles methanol in a chemical group (HC3) including other non-biogenic compounds like ethanol and propane. Thus, individual methanol concentrations are not available and a discussion of the whole chemical group would suffer from the same concerns as for aldehydes. Nevertheless, we agree that methanol is an interesting biogenic compound, so we added the biogenic emissions of HC3 (which solely refer to methanol emissions) and other BVOCs in Section 4.1. We thank the reviewer for pointing this out and hope that we could adapt the manuscript in a sufficient way. We updated Fig.2 of the manuscript (Fig. 2 in this document) and included a new figure showing the additional BVOCs (Fig. 3 in this document) which is now the new Fig.3 in the manuscript. The description of Section 4.1 was modified accordingly and reads now (l.272-305, new count):

*" The effects of model configurations on biogenic emissions of different gases are given in Fig. 2 and Fig 3. As the changes induced by the different model configurations are similar for all presented biogenic gases, the following description focuses on isoprene and HC3 shown in Fig. 2. Note that biogenic HC3 emissions refer solely to methanol which is the only biogenically emitted compound in this chemical group defined in the model.*

*Differences between nighttime (03 UTC) and daytime (09 UTC) emissions are more significant for isoprene than for other biogenic gases. This is because isoprene is a direct product of photosynthesis which is mainly limited to daytime conditions. For the reference setup ("ref"), daytime*

*isoprene emissions are mainly restricted to the Apennine Mountains and two areas within the the central Po valley north of Modena and Bologna. According to USGS land use, these locations are assigned to "Deciduous Broadleaf Forest" and "Crop/Woodland Mosaic", respectively. In contrast to "Dryland Cropland and Pasture" in the rest of the valley, broadleaf trees emit high levels of isoprene. Thus, even small numbers of trees result in significantly increased local isoprene emissions. Biogenic emissions of alpha-pinene, limonene and aldehyde show also increased these regions, but with decreasing characteristic. In contrast, biogenic emissions of methanol and aldehydes almost equally emitted by all vegetation types in this regions. This results in a comparably uniform distribution over most parts of the domain with a significant reduction in the Apennine mountains.*

*The high dependency on tree coverage is emphasized by comparing reference biogenic emissions to emissions based on MODIS land use ("land use"). In contrast to USGS, MODIS does not indicate any trees within the Po valley, which results in negligible biogenic emissions in this region. Although this effect is most prominent for isoprene, significant emission reduction is found for all considered biogenic gases. At the same time, the whole Apennine Mountains and southern foothills of the Alps are assigned to high coverage of broadleaf trees resulting in high isoprene emissions. The use of GFS global meteorology does not change the general emission patterns ("global"). Caused by different initial- and boundary conditions, all biogenic emissions are slightly reduced in the whole region. The implemented response of biogenic emissions to soil dryness significantly influences biogenic emissions ("no SMOIS"). By neglecting this response, emissions are considerably larger than for the reference case, especially in the southern part of the domain. As soil moisture decreases after sunrise, the largest sensitivities are found at 09 UTC for both gases.*

*The RUC LSM induces slightly increased biogenic emissions of all considered gases, in most areas ("LSM"). This general increase is overlapped by a drastic reduction to almost zero emissions in the south-eastern parts of the Po valley for all gases - most prominently visible for biogenic methanol and aldehyde emissions. This reduction is caused by low soil moisture predicted by RUC LSM in the morning hours which results in drought induced plant*

*stress. Using the ACM2 boundary layer- (PBL) and Pleim-Xiu surface layer (SL) schemes instead of MYJ PBL + Eta SL schemes leads to a reduction of biogenic emissions ("PBL + SL"). This effect affects the biogenic emissions of all considered gases and is largest in the eastern central Po valley. Only minor changes in biogenic emissions due to microphysics- and radiation schemes are visible ("microph.", "rad."). Using TGS microphysics instead of the reference WSM6 does only induce small local effects during nighttime (03 UTC). Although being small, effects of using different radiation schemes after sunrise can be attributed to different formulations of shortwave radiation by the Dudhia and RRTMG schemes. "*

Accordingly, a short note on the model variable "HC3" is given in the introduction (l.92, new count):

*Methanol is part of the model variable "HC3" which also includes ethanol and propane as not biogenically emitted alkanes.*

**Additional comments:**

- **l.90: aldehydes are also photochemically produced.**

  Reply: Yes indeed, the reviewer is right. We added this information at the referred position manuscript (l. 93-94, new count):
  *" The model variable "aldehyde" represents a composite of oxidized BVOCs which are affected by several processes including biogenic- and anthropogenic emissions, photochemical production, atmospheric transport and dry deposition. "*

- **l.236+: You might want to refer to Jiang et al., https://doi.org/10.1016/j.atmosenv.2018.01.026 for a discussion of the implementation of drought impact in MEGANv3**

  Reply: Thank you very much for this insightful hint. We are happy to refer to this publication by adding the following sentence in l.258 (new count):
  *" Recently, Jiang et al. (2018) formulate drought response in the subsequent version MEGAN 3 as function of photosynthesis and generalized water stress. "*

**[ACPD](acpd)**

Interactive
comment

- **Figure 4: It is very difficult to pick out the differences highlighted in the discussion in these tiny panels. It would be nice to find another way to illustrate these differences. Perhaps all except the first column should be differences (percent) from the ref case. Or show just averages over the Po Valley and other regions of particular relevance (not on a map).**

  Reply: We agree that the differences are less obvious for friction velocities compared to the other variables shown. We made attempts with the suggested presentation scheme. However, plotting relative difference (i.e. by factors) shown in Fig. 4 of this document may be misleading as they appear to be dominated by small differences of low absolute values at 3 UTC. Thus, showing absolute values provides a visible measure of the magnitude of differences, also in comparison to the other variables. Therefore, we came to the conclusion that the important aspects are more intuitively visible in the original plot, than for factors.

  This is also the case for the newly added surface concentrations of OH in Fig. 8 of the new manuscript.

- **Figures 6 & 7 are also too small - they could at least be enlarged to the width of the page, but difference plots would help illustrate features.**

  Reply: Figures 7 and 8 (new count) have been enlarged which makes it easier to pick individual features.

- **There are a number of spelling and grammar errors, but the paper is understandable.**
  **Here are some corrections: l.112: through → trough**
  **l.269: "neglectable" → negligible**
  **l.305: "friction velocities does only" → "friction velocities only"**
  **l.307: do you mean ?exemplary? (best of its kind) here, or perhaps just "example" or "representative" [similarly elsewhere in the paper]**
  **l.309: does → do**
  **l.353: plain → plane**
  **l.364: hove → have**
  **l.419 "so large" → "to large"**
  **l.420 add comma after "common"**

Reply: We apologize for our trivial mistakes and are grateful for pointing them out. The manuscript was corrected accordingly.

[Figure]

[Figure]

**Fig. 1.** OH surface concentrations on 12 July 2012 at 06 and 09 UTC (coded by colors) for different model configurations.

[Figure]

**Fig. 2.** Isoprene (a) and biogenic HC3 (b) emissions on 12 July 2012 at 03, 06 and 09 UTC (coded by colors) for different model configurations

[Figure]

Fig. 3. Alpha-pinene (a), limonene (b), biogenic ethene (c) and biogenic aldehyde (d) emissions on 12 July 2012 at 03, 06 and 09 UTC (coded by colors) for different model configurations.

[Figure]

**Fig. 4.** Friction velocity factors on 12 July 2012 at 03, 06 and 09 UTC (coded by colors) for different model configurations including increased roughness length ("Z0").

---

## Author Comment (AC2) · 25 Jan 2021

We are grateful for the insightful remarks which lead to substantial improvements of the manuscript. We hope that we could address all concerns in a satisfying way.

General Reply: Thanks to the reviewer's remarks, we see that we were not able to make a key objective sufficiently clear in the manuscript. The mayor concern lies in the missing validation of the meteorological and chemical forecasts by observations. We

definitely see the need for a validation when evaluating the accuracy of such forecasts. However the main objective of this study is meant to be slightly different. This study is not aiming at an optimally implemented case study simulation, with best choices of optional parameterizations, validated by available observations. (This is in contrast to earlier studies of the group, with advanced data assimilation developed and applied, e.g. Elbern et al. 2007: https://doi.org/10.5194/acp-7-3749-2007, Vogel et al. 2020: https://doi.org/10.1016/j.atmosenv.2019.117063.) Here, we aim to show the effects of different model setups on different processes affecting atmospheric distributions of biogenic VOCs (BVOCs) based on only a priori available knowledge, without focus on optimal choices valid for this very situation. Hence, the evaluation of resulting differences does not aim to provide an absolute evaluation which setup performs best in the given case. Instead we aim to point out the pathways of the differences in order to better understand the importance of those. In other words, we want to show the importance of a careful selection of the model setup and the consideration of this aspect when evaluating forecasts e.g. against observations. Identifying mayor pathways of uncertainties from multiple potential error sources induced by the model setup can be seen as a conservative step prior to sufficient model validation by observations.

We are aware of the fact that not all options selected in this study are assumed to be of same accuracy. Nevertheless, the most promising option might not be known before - say - operational simulations and the absolute accuracy of each option differs from case to case. The options used this study are carefully selected and commonly used by the community for "state-of-the-art" simulations.

We revised the manuscript and adopted several formulations in order to clarify this in a satisfying way (abstract I.3-6, introduction I.75-81):

ABSTRACT: " This study identifies and quantifies principal sources of forecasts uncertainties induced by various model configurations under these conditions. Specifically, effects of model configuration on different processes affecting atmospheric distributions of biogenic trace gas distributions are analyzed based on a priori available information. "
INTRODUCTION: " Here, the evaluation does not refer to a quantitative forecast evaluation with respect to observations. Instead we aim to analyze out the pathways of the simulated differences in order to better understand their impact. With this approach, differences in simulated concentrations can be traced back to specific model configuration options affecting different parts of the modeling system. Thus, this study provides a precursory step prior to comprehensive forecast validation by observations as well as probabilistic simulations. Focusing on biogenic gases, various kinds of sensitivities related to model input and setup are considered, including the configuration of the meteorological model. "

In the following, we will reply to the individual aspects of the review step by step:

This paper presents the uncertainties associated with the simulation of some of the biogenic gases depending on the meteorological model settings and landuse datasets. The presented simulation scenarios are focused on the PEGASOS field campaign conducted over the Po valley during summer 2012. Three flight cases are selected for the numerical experiments.

There are large uncertainties in simulating the fluxes of the biogenic volatile organic compounds and their mixing ratios in the air quality models. It's worth studying these uncertainties. However, my primary concern here is that the authors run the WRF model with different input data and physics schemes without verifying how suitable are the selected model settings for the given task. Here, the role of the WRF model is to provide the meteorological drivers to the EURAD-IM offline chemistry transport model. The authors don't provide any verification of the model simulations using the surface or aircraft observations. It's likely that some of the presented model scenarios aren't able to accurately simulate the meteorology in the Po valley for the selected days.

Reply1: We agree with the reviewer that different meteorological forecasts provided by  $\overline{\text{WRF}}$  might induce significant errors to the chemical forecasts in the Po valley. In fact,

**ACPD**
one of the main objectives of this paper is to show the importance of these differences. The meteorological uncertainties provide one type of potential uncertainties, the different effects of which are investigated in this study. Referring to our general reply above, we aim to achieve this by showing the differences between those and not proving an absolute evaluation. The setups of WRF used here have been used in several other studies which analyze their performances.

Nevertheless, we see the advantage of a basic validation of the meteorological forecast provided by WRF for describing the meteorological conditions. Low level meteorological conditions from the simulations are compared with radiosonde observations at San Pietro Capofiume (SPC, central Po valley) perfromed by the Italian Meteorological Service at 00 UTC. The following section was added in Section 2.1 of the (I.130-137, new count):

" Radiosonde observations at San Pietro Capofiume (SPC) from the Italian Meteorological Service are launched at 00 UTC each day. On 12 July 2012, the sounding states calm conditions with westerly winds of about 1.5  $m s^{-1}$  to 5  $m s^{-1}$  in the lowest 100 m. The temperature at 00 UTC was  $21.2^{\circ}C$  close to the surface with a temperature inversion reaching up to  $26.8^{\circ}C$  in 200 m height. At this time, the relative humidity reduces steadily from 69% to 42% in about 1 km height. These local conditions are sufficiently well simulated by the WRF forecasts for all model configurations used in this study at 00 UTC. Simulated surface near winds are between  $2.5 m s^{-1}$  and  $5.5 m s^{-1}$  from south-western directions, relative humidity close to the surface ranges from 50% to 69% and temperatures vary between  $21.3^{\circ}C$  and  $24.6^{\circ}C$ . All simulations capture the inversion, but tend to underestimate its intensity as maximum temperatures in 200 m height are about  $25^{\circ}C$ . "

Also, accurate simulation of the soil moisture in WRF usually requires model spin-up over some time period, so using a "good" land-surface scheme isn't sufficient. Thus, using inaccurate or unverified meteorological simulations to drive the offline EURAD-IM chemistry model doesn't make sense.
Reply2: We agree that spin up is essential especially for soil moisture which might have long memory. We accounted for this in two steps: Firstly, all WRF simulations used in this study are initialized by global meteorological fields provided by ECMWF and NOAA. As both global models are operationally used, we rely on the high resolution global fields assuming that soil moisture is physically consistent in their analyses. Secondly, we are aware that fields of soil moisture still require some spin up when using these global fields in WRF with different soil hydrology. We therefore initialized all the simulations (WRF and EURAD-IM) 27 hours before the first time of evaluation. At this time, initial differences from the global fields have vanished which indicates a reasonable spin-up period in this case. Consequently, surface skin soil moisture from both surface schemes used in this study show a clear diurnal cycle with a long term trend towards surface dry out, which is expected under precipitation-free conditions during this case. As well known from NWP, initializing a longer model forecast without observational correction leads again to increasing deviation of the simulated fields from reality.

Based on this, we came to the conclusion that potential remaining errors related to soil moisture spin up are small compared to the sensitivities we are investigating in this study. We thank the reviewer for highlighting this important aspect. We added related details in the general model description (I. 160-163, new count):

" For all selected cases, the simulations of WRF and EURAD-IM are initialized one day before the day of evaluation at 00 UTC, each. Thus, at least 27 hours of spin-up are performed for all meteorological and chemical fields in addition to the initialization of meteorological fields including soil moisture in WRF."

One of the selected sensitivity simulations is done using the MODIS versus USGS land-use dataset. It's expected that the differences in the vegetation
map for the Po valley will lead to large differences in the fluxes of the biogenic VOCs. However, the MODIS data is more up-to-date than the USGS land-use data. Therefore, it isn't clear what we learn by testing both meteorological and air quality models using the old (probably not accurate) land-use data (USGS) as input.

Reply3: Indeed, MODIS can be expected to provide more detailed information than USGS land use data. Although this has been noted in the manuscript (I.200-217, old count), we agree that it should have been made clearer, why USGS is still used in this study. USGS has been successfully used in a great number of studies over the last decades and is still a commonly used database in WRF and also other NWP systems. Moreover, in order to investigate the effects of uncertainties in land use information, (at least) two different data bases are required. Comparing MODIS information with USGS data offers the ability to make use of two generally different - yet realistic data sets. This provides more reasonable results than e.g. modifying MODIS information artificially. We adopted the description of the model inputs accordingly and hope that we could explain this point sufficiently well (I.227-231, new count):

" Multiple studies indicate a more detailed and reliable characterization compared to AVHRR based products (e.g., Hansen et al., 2002; Smirnova et al., 2016). However, GLCC land use information provide a basically different data set which has been used in a great number of studies over the last decades until today (e.g., Krinner et al., 2005; Dee et al., 2011; Sellar et al., 2019). Thus, using GLCC data from USGS as reference option provides a solid basis for investigating changes in land use information induced by MODIS. "

The analysis of the effects of source regions (section 4.3.2) is interesting, but again without the verification of the wind speed/direction and other meteorological variables, it's impossible to determine which model simulations or scenarios are realistic here. It's possible that some of the WRF model scenarios are somewhat similar in terms of their forecast skill, but at least basic model verification
**is required to select such model configurations to conduct reasonable meteorological simulations to be used as input in the chemistry transport model.**

Reply4: This aspect relates to a previous concern, which has been addressed in Reply1.

To the meteorological validation with radiosonde observations in Section 2.1 (compare Reply1) is also referred in the description of the source regions in Section 4.3.2 (I.347-350, new count):

" Although differences in friction velocities do not vary substantially in this case, an investigation of the airmasses' history provides more detailed information on pollutant transport. The analysis of the meteorological conditions states the overall reproducibility of the observed local conditions by all simulations used in this study (Sect. 2.1). At the same time, the simulations show low level variable wind directions related to the low wind speeds which might result in diverging pollutant transport. "

The simulation of other terpenes (e.g. alpha-pinene) isn't presented here. Instead, the authors present the simulation of the aldehydes, which are also produced by the gas chemistry in the EURAD-IM model. This aspect requires additional analysis.

Reply5: We totally agree with the reviewer that aldehydes are affected by a large number of processes like secondary production. Individual effects of those cannot be separated by the approach used in this study. We therefore decided to withdraw the analysis of aldehyde concentrations from Section 5 and noted this issue in the beginning of this section (I.386-389, new count) as follows:

" The evaluation of biogenic gas concentrations is restricted to isoprene because of its direct dependency on the model processes discussed above. Other biogenic gases are affected by additional processes like secondary production which hamper a detailed evaluation. Instead, resulting OH concentrations are analyzed on their impact on reactive atmospheric chemistry."

ACPD
Additionally, we added a set of other BVOC including terpenes to the discussion of biogenic emissions in Section 4.1. Fig. 2 and 3 of the manuscript show now biogenic emissions from isoprene, HC3 (consisting only of biogenic methanol emissions) as well as alpha-pinene, limonene, biogenic ethene and biogenic aldehydes. As uncertainties in these gases appear to be highly similar, a detailed discussion is only provided for isoprene and methanol. Methanol is selected for detailed discussion as it is the second abundant gas in this case and shows differing absolute distributions compared to isoprene. We thank the reviewer for pointing towards these important biogenic gases and hope that we could adapt the manuscript in a satisfying way. We updated Fig.2 of the manuscript (Fig. 1 in this document) and included a new figure showing the additional BVOCs (Fig. 2 in this document) which is now the new Fig.3 in the manuscript. The description of Section 4.1 was modified accordingly and reads now (I.272-305, new count):

" The effects of model configurations on biogenic emissions of different gases are given in Fig. 2 and Fig 3. As the changes induced by the different model configurations are similar for all presented biogenic gases, the following description focuses on isoprene and HC3 shown in Fig. 2. Note that biogenic HC3 emissions refer solely to methanol which is the only biogenically emitted compound in this chemical group defined in the model.

Differences between nighttime (03 UTC) and daytime (09 UTC) emissions are more significant for isoprene than for other biogenic gases. This is because isoprene is a direct product of photosynthesis which is mainly limited to daytime conditions. For the reference setup ("ref"), daytime isoprene emissions are mainly restricted to the Apennine Mountains and two areas within the the central Po valley north of Modena and Bologna. According to USGS land use, these locations are assigned to "Deciduous Broadleaf Forest" and "Crop/Woodland Mosaic", respectively. In contrast to "Dryland Cropland and Pasture" in the rest of the valley, broadleaf trees emit high levels of isoprene. Thus, even small numbers of trees result in significantly increased local isoprene emissions. Biogenic emissions of alpha-pinene, limonene and aldehyde show also increased these regions, but with decreasing characteristic. In contrast, biogenic emissions of
methanol and aldehydes almost equally emitted by all vegetation types in this regions. This results in a comparably uniform distribution over most parts of the domain with a significant reduction in the Apennine mountains.

The high dependency on tree coverage is emphasized by comparing reference biogenic emissions to emissions based on MODIS land use ("land use"). In contrast to USGS, MODIS does not indicate any trees within the Po valley, which results in negligible biogenic emissions in this region. Although this effect is most prominent for isoprene, significant emission reduction is found for all considered biogenic gases. At the same time, the whole Apennine Mountains and southern foothills of the Alps are assigned to high coverage of broadleaf trees resulting in high isoprene emissions. The use of GFS global meteorology does not change the general emission patterns ("global"). Caused by different initial- and boundary conditions, all biogenic emissions are slightly reduced in the whole region. The implemented response of biogenic emissions to soil dryness significantly influences biogenic emissions ("no SMOIS"). By neglecting this response, emissions are considerably larger than for the reference case, especially in the southern part of the domain. As soil moisture decreases after sunrise, the largest sensitivities are found at 09 UTC for both gases.

The RUC LSM induces slightly increased biogenic emissions of all considered gases, in most areas ("LSM"). This general increase is overlapped by a drastic reduction to almost zero emissions in the south-eastern parts of the Po valley for all gases - most prominently visible for biogenic methanol and aldehyde emissions. This reduction is caused by low soil moisture predicted by RUC LSM in the morning hours which results in drought induced plant stress. Using the ACM2 boundary layer- (PBL) and Pleim-Xiu surface layer (SL) schemes instead of MYJ PBL + Eta SL schemes leads to a reduction of biogenic emissions ("PBL + SL"). This effect affects the biogenic emissions of all considered gases and is largest in the eastern central Po valley. Only minor changes in biogenic emissions due to microphysics-and radiation schemes are visible ("microph.", "rad."). Using TGS microphysics instead of the reference WSM6 does only induce small local effects during nighttime (03 UTC). Although being small, effects of using different radiation schemes after sunrise can be attributed to
different formulations of shortwave radiation by the Dudhia and RRTMG schemes. "

Accordingly, a short note on the model variable "HC3" is given in the introduction (I.92, new count):

" Methanol is part of the model variable "HC3" which also includes ethanol and propane as not biogenically emitted alkanes. "

**Again, it's hard to make any conclusions with respect to the accuracy of the gas chemistry simulations as none of the simulated chemical species are compared against the aircraft or other measurements.**

Reply:6 This aspect refers to the initial concern, which has been addressed in the general reply. We hope, that we could explain our objective sufficiently in the reply as well as in the manuscript.

**The discussion of the dry deposition is interesting, however, the role of more important processes such as photochemistry is necessary to consider.**

Reply7: We are thankful for this note. In this context, OH provides useful insights in relation to isoprene concentrations and downstream photochemistry. Therefore, we added an analysis of OH surface concentrations including Fig. 8 (Fig. 3 in this document) and Subsection 5.2 (I. 406-419, new count) in the manuscript (replacing aldehyde concentrations, as described above).

" The hydroxy radical OH is a highly reactive oxidant in the atmosphere acting as most important sink of isoprene (Kaser et al., 2015). Generally, OH may be influenced by the model configuration via reaction with biogenically emitted gases and meteorological conditions. Local meteorology mainly affects OH by changes in radiation related to humidity and clouds. In this specific case, the weather in the Po region was continuously characterized by clear and dry Interactive comment

conditions as described in Sect. 2.1. Thus, no significant differences in humidity and cloud coverage are simulated by the model configurations (not shown). This renders the differences in OH concentrations being determined by changed biogenic VOCs. "

As expected from atmospheric chemistry, daytime OH concentrations shown in Fig. 8 are reduced in regions of high BVOC concentrations like the central Po valley and the southern Apennines. In contrast, OH concentrations remain comparably high in the mountains and over the ocean were isoprene concentrations are neglectable. This direct dependence of OH to biogenic gases causes also significant differences in OH concentrations with respect to model configuration. In this case, the effects are most dominant in cases of increased isoprene concentrations with respect to the reference simulation. Significant reduction of OH is induced by excluding drought response ("no SMOIS"), RUC LSM, and less pronounced for MODIS land use in the southern Apennines. While these reductions are persistent in time, increased isoprene concentrations in the central Po valley for GFS global meteorology at 06 UTC result in temporally reduced OH concentrations in this region. "

Related modifications have been made in the conclusions (I.467-470, new count) as well as the abstract (I.12-14, new count):

ABSTRACT: " As a result, large sensitivities to model configuration are found for surface concentrations of isoprene as well as OH, affecting reactive atmospheric chemistry. "

CONCLUSIONS:" Moreover, changes in surface concentrations of biogenic trace gases induce significant differences in OH concentrations affecting reactive atmospheric chemistry. Excluding the emission response to drought stress reduces local OH concentrations by up to a factor of three in this study. "

Based on the aforementioned shortcomings of the study, I urge the authors to redo the only difference induced by EBIO by definition) Interactive numerical experiments, conduct extensive model verifications, and submit a substantially
**revised version of the paper in the future.**

Final Reply: The concerns described in the review allowed us to apply substantial modifications to the manuscript - especially with respect to the understandability of the overall objectives and the traceability of the specific setup. In our opinion, these modifications lead to essential improvements of the manuscript. We hope that this is in accordance with the reviewers opinion.
Interactive

comment